# The *Caenorhabditis elegans* homolog of the *Evi1* proto-oncogene, *egl-43*, coordinates G1 cell cycle arrest with pro-invasive gene expression during anchor cell invasion

Ting Deng[1,2], Przemyslaw Stempor[3], Alex Appert[3], Michael Daube[1], Julie Ahringer[3], Alex Hajnal[1]*, Evelyn Lattmann[1]

1 Institute of Molecular Life Sciences, University of Zurich, Winterthurerstrasse, Zürich, Switzerland, 2 Molecular Life Science PhD Program, University and ETH Zurich, Zürich, 3 The Gurdon Institute and Department of Genetics, University of Cambridge, Cambridge, United Kingdom

* alex.hajnal@mls.uzh.ch

**Data Availability Statement:** All relevant data are within the manuscript and its Supporting Information files. The ChIP-seq data generated in

## Abstract

Cell invasion allows cells to migrate across compartment boundaries formed by basement membranes. Aberrant cell invasion is a first step during the formation of metastases by malignant cancer cells. Anchor cell (AC) invasion in *C. elegans* is an excellent *in vivo* model to study the regulation of cell invasion during development. Here, we have examined the function of *egl-43*, the homolog of the human *Evi1* proto-oncogene (also called *MECOM*), in the invading AC. *egl-43* plays a dual role in this process, firstly by imposing a G1 cell cycle arrest to prevent AC proliferation, and secondly, by activating pro-invasive gene expression. We have identified the AP-1 transcription factor *fos-1* and the *Notch* homolog *lin-12* as critical *egl-43* targets. A positive feedback loop between *fos-1* and *egl-43* induces pro-invasive gene expression in the AC, while repression of *lin-12 Notch* expression by *egl-43* ensures the G1 cell cycle arrest necessary for invasion. Reducing *lin-12* levels in *egl-43* depleted animals restored the G1 arrest, while hyperactivation of *lin-12* signaling in the differentiated AC was sufficient to induce proliferation. Taken together, our data have identified *egl-43 Evi1* as an important factor coordinating cell invasion with cell cycle arrest.

## Author summary

Cells invasion is a fundamental biological process that allows cells to cross compartment boundaries and migrate to new locations. Aberrant cell invasion is a first step during the formation of metastases by malignant cancer cells. We have investigated how a specialized cell in the Nematode *C. elegans*, the so-called anchor cell, can invade into the adjacent epithelium during normal development. Our work has identified an oncogenic transcription factor that controls the expression of specific target genes necessary for cell invasion, and at the same time inhibits the proliferation of the invading anchor cell. These findings shed light on the mechanisms, by which cells decide whether to proliferate or invade.

this study are available at the NCBI Gene Expression Omnibus (GEO) (http://www.ncbi.nlm.nih.gov/geo/) under accession number GSE144292.

**Funding:** This work was supported by the Swiss National Science Foundation grant no. 31003A-166580 and by the Kanton of Zürich. Julie Ahringer's laboratory was supported by Wellcome grant 101863 and by core funding from the Wellcome Trust (092096) and Cancer Research UK (C6946/A14492). The funders had no role in study design, data collection and analysis, decision to publish, or preparation of the manuscript.

**Competing interests:** The authors have declared that no competing interests exist.

## Introduction

Cell invasion, which is initiated by the breaching of basement membranes (BMs), is a regulated physiological process allowing select cells to cross compartment boundaries during normal development. Cell invasion is also the first step that is activated during the formation of metastases by malignant cancer cells [1]. Anchor cell (AC) invasion in *C. elegans* is a genetically amenable and tractable model that has provided important insights into the molecular pathways regulating cell invasion and uncovered the molecular similarities between tumor cell and developmental cell invasion [2,3].

The AC is specified during the second larval stage (L2) of *C. elegans* development, when two equivalent precursor cells (Z1.ppp and Z4.aaa) adopt either the AC or the ventral uterine (VU) fate, depending on stochastic differences in LAG-2 Delta/ LIN-12 Notch signaling [4,5]. This initially small difference is amplified by two extra stochastic events, the division order of Z1 and Z4 and the expression onset of *hlh-2* in Z1.pp and Z1.aa[6]. The cell that exhibits higher *lag-2* expression levels adopts the "default" AC fate and activates LIN-12 Notch signaling in the adjacent cell to induce the VU fate [7]. Unlike the VU cells, which undergo three to four rounds of cell divisions, the AC never divides but remains arrested in G1 phase and adopts an invasive fate. NHR-67, a nuclear receptor of the *tailless* family, is required to maintain the G1 arrest of the AC by regulating the cyclin-dependent kinases (CDK) inhibitor CKI-1. In response to the G1 arrest established by NHR-67, the histone deacetylase HDA-1 promotes the invasive AC fate [8]. AC invasion occurs during the third larval stage (L3) of *C. elegans* development. During this process, the AC is guided ventrally by cues from the adjacent primary vulva precursor cells (VPCs) and the ventral nerve cord. The AC then breaches the two BM layers separating the uterus from the epidermis and establishes direct contact with the invaginating vulva epithelium [9].

Both AC/VU fate specification and AC invasion require the activity of the e*gl-43* gene, which encodes two isoforms of a Zinc finger transcription factor homologous to the mammalian *Evi1* proto-oncogene in the MECOM (Myelodysplastic Syndrome 1(MDS1) and Ecotropic Viral Integration Site 1 (EVI1), MDS1-EVI1) locus [10–12]. Human *Evi1* has been implicated in the development of different types of cancer, most notably in the hematopoietic system in acute myeloid leukemia (AML) [13,14]. Inhibition of *C. elegans egl-43* during the mid-L2 stage leads to the formation of two ACs due to a defect in AC/VU cell specification [10]. However, *egl-43* remains expressed in the AC after its specification, where EGL-43 is required together with the AP-1 transcription factor FOS-1 to induce the expression of genes that promote BM breaching (i.e. pro-invasive genes), such as *zmp-1*, *cdh-3* and *him-4* [10,11,15].

Despite the importance of *egl-43* in regulating pro-invasive gene expression and BM breaching, its exact role during AC invasion remains poorly understood. Here, we report that *egl-43* and *fos-1* form a positive auto-regulatory feedback-loop. Moreover, *egl-43* plays a previously unknown role in establishing the G1 arrest of the AC by repressing Notch-dependent AC proliferation. Thus, *egl-43* acts as an important regulator of AC invasion that coordinates the G1 arrest with pro-invasive gene expression.

## Results

### Deletion of the long *egl-43L* isoform leads to the formation of multiple ACs that fail to invade

In order to study the role of the different *egl-43* isoforms during AC invasion, we used CRISPR/Cas9 genome editing [16] to insert *gfp* tags at the 5' or 3' end of the *egl-43* locus (**Fig 1A**). Since the transcriptional start site of the short isoform (*egl-43S*) is located within the 5th

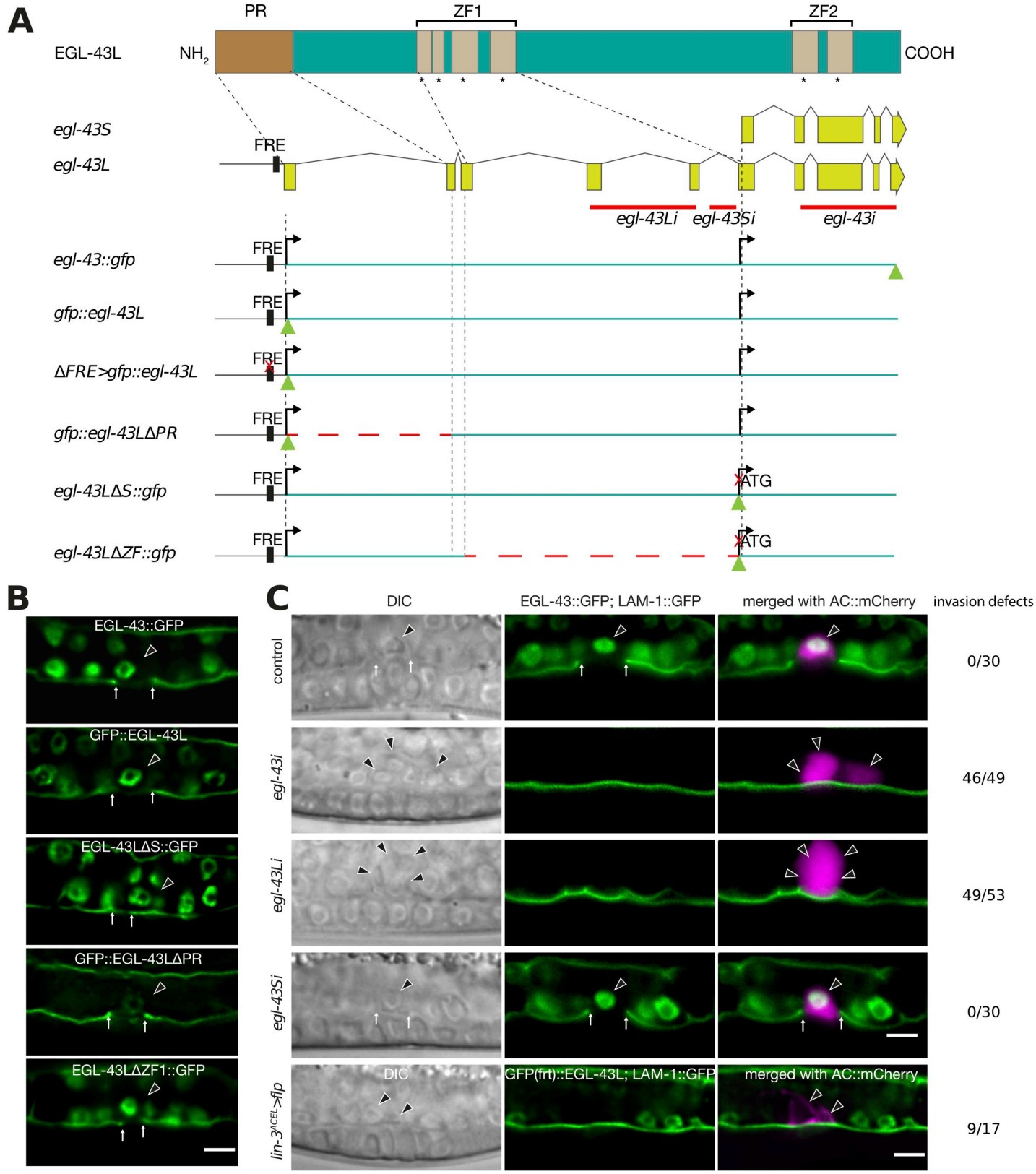

**Fig 1. Loss of *egl-43L* function leads to the formation of multiple ACs. (A)** Schematic overview of the *egl-43* locus and the CRISPR/Cas9 engineered alleles used in this study. Green triangles indicate *gfp* insertion sites and dashed red line indicate deleted regions. The red crosses indicate the sites of the 11 bp (TTACTCATCTT)

deletion in the Fos-Responsive Element (FRE) and of the mutation in the initiation codon of the *egl-43S* isoform. Solid red lines refer to the regions targeted by RNAi. **(B)** Expression patterns of the different endogenous *egl-43* reporters depicted in **(A)**. The basement membrane (BM) was simultaneously labelled with the LAM-1::GFP marker to score AC invasion. At least 25 animals were examined for each reporter. All animals contained one GFP expressing AC and exhibited BM breaching. A quantification of the AC expression levels is shown in **S2 Fig**. **(C)** RNAi of the different *egl-43* isoforms and FLP/FRT-mediated excision of *egl-43L*. Left panels show Nomarski (DIC) images and middle panels show the fluorescence signals of endogenous EGL-43::GFP expression in the nuclei and the LAM-1::GFP BM marker. The right panels show the GFP signals merged with the ACs labelled by the *lin-3^ACEL^>mCherry* reporter (rows 1–4) and *cdh-3>mCherry::PH* (row 5) in magenta. The black arrowheads point at the AC nuclei and the white arrows at the locations of the BM breaches. Control refers to animals exposed to the empty RNAi vector. The numbers to the right indicate the penetrance of the invasion defects. A quantification of the AC expression levels is shown in **S2C Fig**. The scale bars are 5 μm.

intron of the *egl-43L* locus, the insertion of the *gfp* cassette at the 5' end of the first exon labels exclusively the protein encoded by the long isoform (*gfp::egl-43L*), whereas the insertion of the *gfp* tag at the 3' end of the locus labels both, the short *egl-43S* and the long *egl-43L* isoforms (*egl-43::gfp*, **Fig 1A**). Furthermore, the *gfp* cassette inserted at the 5' end contained two flippase recognition target (FRT) sites in the *gfp* introns, permitting the tissue-specific inactivation of the *egl-43L* isoform (**S1 Fig**). For the expression analysis, we focused on the mid-L3 stage (the Pn.pxx stage, after the VPCs had undergone two rounds of cell divisions), the time when AC invasion normally occurs [9]. Both reporters, *gfp::egl-43L* and *egl-43::gfp*, were expressed in the invading AC and in the surrounding ventral and dorsal uterine (VU and DU) cells (**Fig 1B**, for a quantification of the AC expression levels of the different reporters, see **S2 Fig**). We found no obvious difference in the expression pattern of the two *egl-43* reporters, suggesting that the long *egl-43L* isoform accounts for most of the expression observed. In order to test if the uterine expression is indeed caused by *egl-43L*, we designed two RNAi clones, one specifically targeting *egl-43L* (exons 4 and 5 of *egl-43L*), and the other targeting the 5' UTR of *egl-43S*, which is not included in the *egl-43L* transcript (**Fig 1A**). EGL-43::GFP expression was lost upon *egl-43L* RNAi, yet no difference in expression was observed after *egl-43S* RNAi (**Fig 1C, S2C Fig**), supporting the above conclusion that the uterine expression originates predominantly from the long *egl-43L* isoform.

As reported previously [11], *egl-43L* RNAi led to an invasion defect with a penetrance comparable to that of total *egl-43* RNAi (92% for *egl-43L* (n = 53) and 94% for total *egl-43* RNAi (n = 49), combined data from two independent RNAi experiments). Consistent with a role of *egl-43* during AC/VU specification [10,11], we detected early L3 larvae with two ACs upon *egl-43L* or total *egl-43* RNAi (**Fig 1C**). However, in 36 out of 49 cases, total *egl-43* RNAi led to the formation of more than two ACs, a phenotype that cannot be explained by an AC/VU specification defect (**Fig 1C**). Similar to the invasion defects, the occurrence of multiple ACs was also caused by selective inhibition of the long *egl-43L* isoform, with 42 out of 53 worms containing more than 2 ACs (**Fig 1C**). Moreover, the number of ACs increased progressively with the age of the larvae, indicating an ongoing proliferation of the AC after the L2 stage (**Fig 2A and 2B**). This pointed to an additional role of *egl-43L* in preventing the proliferation of the AC after its specification, independently of its function during VU fate specification.

In order to specifically examine the role of the *egl-43S* isoform during AC invasion, we used CRISPR/Cas9 engineering to introduce an ATG→CTG mutation in the predicted *egl-43S* start codon (*egl-43LΔS::gfp*, **Fig 1A and 1B**). The *egl-43LΔS::gfp* strain was viable and showed a similar expression pattern as the *egl-43::gfp* and *gfp::egl-43L* strains (**Fig 1B**). Moreover, AC invasion occurred normally in all *egl-43LΔS::gfp* animals examined, and no AC proliferation was observed (n = 35), which confirms the predominant role of *egl-43L* in regulating AC invasion and proliferation.

Finally, we used the Flp/FRT system to generate an AC-specific knock-out of *egl-43L* [17]. Since two FRT sites had been inserted in introns of the *gfp* cassette at the 5' end of the *egl-43L* locus, the expression of the FLPase under control of the AC-specific *lin-3* enhancer element

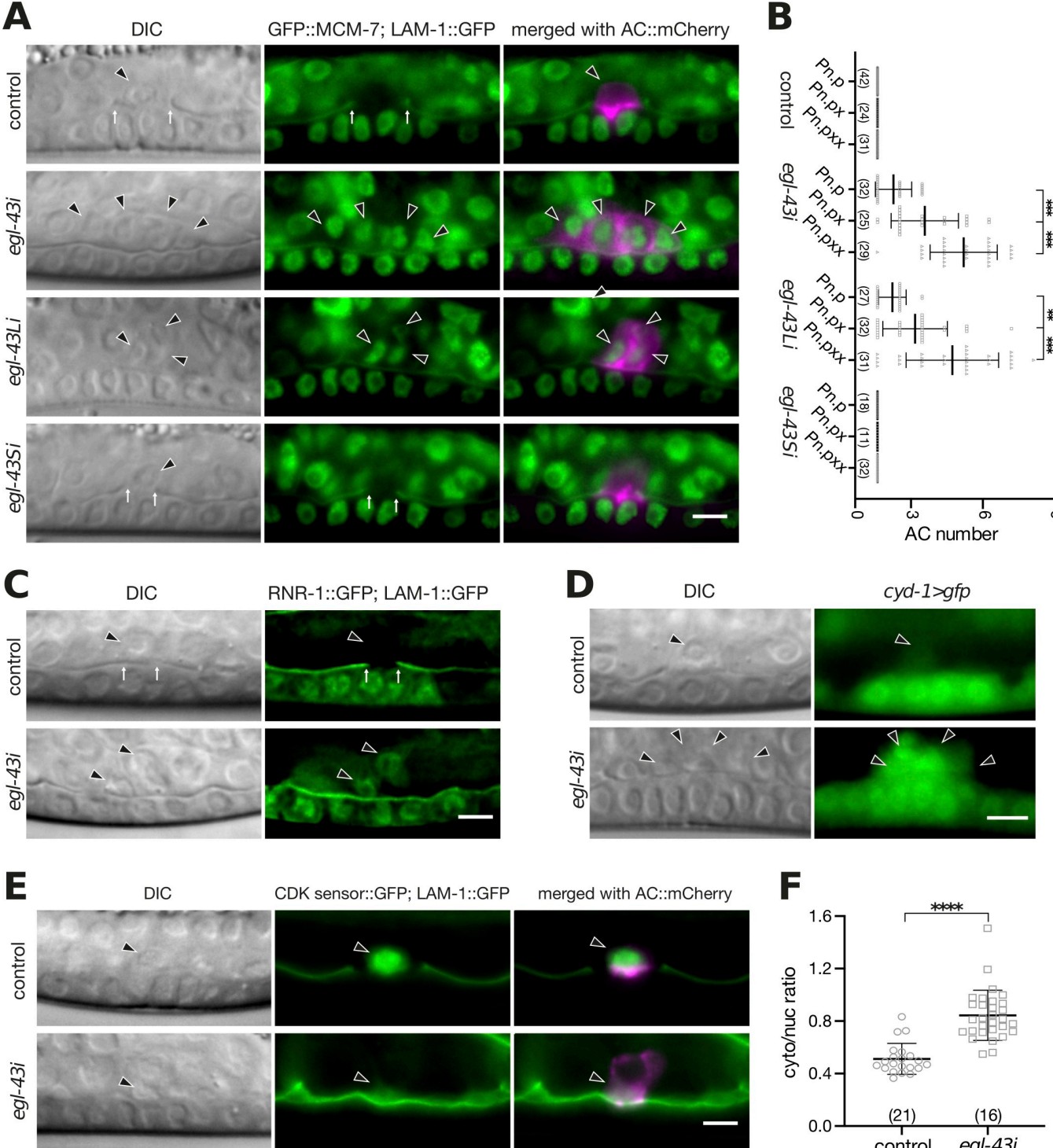

**Fig 2. *egl-43* is required for the G1 arrest of the AC. (A)** Expression of the endogenous GFP::MCM-7 reporter after RNAi of the different *egl-43* isoforms. Elevated AC expression of GFP::MCM-7 was detected after *egl-43* RNAi and *egl-43L* RNAi (29/29 and 28/31 cases, respectively), but not after control or *egl-43S* RNAi (0/31 and 0/32 cases, respectively). The left panels depict Nomarski (DIC) images and the middle panels nuclear GFP::MCM-7 expression together with the LAM-1::GFP BM marker at the mid-L3 stage. The right panels show the GFP signals merged with the ACs labelled by the *cdh-3*>mCherry::PH reporter in magenta. **(B)** Quantification of the number of ACs formed after RNAi of the different *egl-43* isoforms from the early L3 (Pn.p) until the mid-L3 (Pn.pxx) stage. The error bars indicate the standard deviation. **(C)** Expression of the S-phase marker RNR-1::GFP together with LAM-1::GFP in control and *egl-43* RNAi-treated mid-L3 larvae. No RNR-1::GFP expression was detected in 48 control RNAi animals, while 30/39 *egl-43* RNAi- treated animals expressed the S-phase marker. **(D)** Elevated expression of the Cyclin D *cyd-1*>*gfp* reporter in the multiple ACs of *egl-43* RNAi-treated mid-L3 larvae was observed in 52/58 cases,

while all 43 control larvae showed faint *cyd-1>gfp* expression in the single AC. **(E)** Expression of the CDK activity sensor in the AC of control and *egl-43* RNAi-treated mid-L3 larvae together with the LAM-1::GFP BM marker. The right panels show the CDK sensor signal merged with the ACs labelled by the *lin-3^{ACEL}>mCherry* reporter in magenta. **(F)** Quantification of CDK sensor activity in the AC. Scatter plots showing the cytoplasmic to nuclear intensity ratio as a measure of CDK activity. The error bars indicate the standard deviation and the vertical bars in **(B)** and horizontal bars in **(F)** the mean values. Statistical significance was determined with a Student's t-test and is indicated with ** for p<0.01, *** for p<0.001, **** for p<0.0001 and n.s for p> 0.05. In each graph, the numbers of animals scored are indicated in brackets. The black arrowheads point at the AC nuclei and the white arrows at the locations of the BM breaches. The scale bars are 5 μm.

(*lin-3^{ACEL}>flp*) [18] specifically inactivated *egl-43L* in the AC (**S1 Fig**). Deletion of *egl-43L* in the AC led to invasion defects (9 out of 17 animals) and the formation of multiple ACs (4 out of 17 cases), similar to the phenotypes caused by *egl-43* RNAi (**Fig 1C**). The relatively low penetrance observed after Flp/FRT-mediated excision compared to RNAi could be due to the perdurance of the EGL-43 protein in the AC, as faint GFP::EGL-43 expression in the AC of mid-L3 larvae could be observed in 2 out of 17 cases.

In summary, the long *egl-43L* isoform acts cell-autonomously to block AC proliferation and promote invasion.

## Neither the N-terminal PR nor the ZF1 domains in EGL-43 are necessary for AC invasion

*egl-43L* encodes a transcription factor containing an N-terminal PRD1-BF1/RIZ-1 domain (PR) and two separate clusters of Zinc finger domain (ZF1 & ZF2) that could exhibit DNA binding activity [12]. PR domains are structurally similar to the SET (Su(var)3-9, Enhancer of zeste and Trithorax) domains, which contribute to histone lysine methyltransferase activity and have also been reported to mediate protein-protein interactions [19,20]. In order to dissect the requirement of the different EGL-43 domains during AC invasion, we generated in-frame deletions in the *egl-43* coding region using the CRISPR/Cas9 system (**Fig 1A**). In the PR domain deletion allele (*gfp::egl-43ΔPR*, deleted amino acids 2–62), we detected an approximately 75% decrease in GFP::EGL-43L expression levels in the AC (**Fig 1B**, **S2A Fig**). Despite this strong reduction, neither the proliferation nor the invasion of the AC were affected as all 25 animals scored showed normal AC invasion and no proliferation. Thus, the PR domain in EGL-43 is not necessary for AC invasion. Though, the reduced expression levels suggested that the PR domain is either required for *egl-43* autoregulation (see below), protein stability, or that there exist additional regulatory elements in the deleted first intron that promote *egl-43* expression in the AC (**Fig 1A**).

Furthermore, no AC invasion defects and no change in expression levels were observed in *egl-43ΔZF1::gfp* mutants, which carry an in-frame deletion removing amino acids 161–235, which encode the N-terminal Zinc Finger domains (ZF1) (**Fig 1B**, **S2B Fig**). Note that this allele also deletes the promoter of the *egl-43S* isoform. By contrast, the *egl-43* null mutant *tm1802*, which contains a 659 bp deletion removing the C-terminal Zinc Finger domains (ZF2), displays severe developmental defects and early embryonic lethality [11]. Thus, the regulation of AC invasion and proliferation most likely depends on the C-terminal ZF2 domains.

## *egl-43* is required for the G1 arrest of the AC

Previous studies have shown that the AC must remain arrested in the G1 phase in order to invade [8]. The occurrence of multiple (i.e. more than two) ACs upon inactivation of *egl-43L* indicated that the AC had bypassed the G1 arrest and begun to proliferate. In order to test this hypothesis, we performed RNAi knock down of the different *egl-43* isoforms in a strain carrying an endogenous GFP::MCM-7 reporter. *mcm-7* encodes a subunit of the pre-replication complex (pre-RC) that is highly expressed in cycling cells but down-regulated in non-proliferating cells [21,22]. No GFP::MCM-7 expression was detectable in the AC after control or *egl-*

*43S* RNAi. However, *egl-43* and *egl-43L* RNAi resulted in elevated GFP::MCM-7 levels in the multiple ACs that formed, indicating that these ACs had re-entered the cell cycle (**Fig 2A**). Consistent with this conclusion, the average number of ACs increased during the development from the Pn.p (late L2/early L3) stage to the Pn.pxx (mid to late L3) stage (**Fig 2B**).

To further characterize the cell cycle state of the AC, we analyzed RNR-1::GFP expression, which serves as an S-phase marker [23]. While RNR-1::GFP was absent in the AC of control RNAi animals, it was expressed in the multiple ACs formed after *egl-43* RNAi (**Fig 2C**). Moreover, we detected elevated levels of a transcriptional *cyd-1>gfp* Cyclin D reporter in the multiple ACs produced after *egl-43i*, while the single AC in control animals showed only faint Cyclin D expression (**Fig 2D**). Finally, we expressed a CDK biosensor in the AC to directly quantify CDK kinase activity [24,25]. This sensor monitors CDK activity via the phosphorylation-induced nuclear export of a GFP-tagged kinase substrate. Thus, a high cytoplasmic-to-nuclear signal ratio indicates high CDK activity. The multiple ACs formed in *egl-43* RNAi-treated animals showed a significantly increased CDK sensor activity compared to the single AC in control RNAi animals (**Fig 2E and 2F**).

Thus, loss of EGL-43 function increases CDK activity and triggers cell cycle entry of the AC.

## *egl-43* and *nhr-67* play distinct roles in inducing the G1 arrest of the AC

As reported previously [8], RNAi of *nhr-67* led to the appearance of multiple ACs that could not breach the BM (**Fig 3A**). Since loss of *nhr-67* led to a similar phenotype as inhibition of *egl-43*, we tested whether the expression of *nhr-67* depends on *egl-43*, or vice versa. We first measured *egl-43* and *nhr-67* GFP reporter expression in early L3 larvae at the Pn.p stage, shortly after AC specification had occurred and the G1 arrest had been established. Quantification of endogenous GFP::EGL-43L reporter levels after *nhr-67* RNAi revealed no significant change (**S3A and S3B Fig**). Also, the expression of an *nhr-67::gfp* reporter at the early L3 stage was not significantly changed by *egl-43* RNAi (**Fig 3C and 3D**). Moreover, the expression of the histone deacetylase *hda-1*, which acts downstream of *nhr-67* to promote AC invasion [8], was not changed by *egl-43* RNAi (**S3E and S3F Fig**). However, in mid-L3 larvae (Pn.pxx stage), around the time of AC invasion, *nhr-67* RNAi caused an approximately 30% reduction in GFP::EGL-43 expression levels, and a similar reduction in NHR-67::GFP expression levels was observed after *egl-43* RNAi (**Fig 3A–3D**). Thus, at later stages *egl-43* and *nhr-67* may positively regulate each other's expression.

Due to the high penetrance of the AC proliferation phenotype caused by single *nhr-67* or *egl-43* RNAi, we were unable to test if the simultaneous knock-down of both transcription factors had an additive effect. Consistent with a previous report [8], the AC-specific expression of the CDK inhibitor *cki-1* together with an mNeonGreen (mNG) marker on a bi-cistronic mRNA under control of a *cdh-3* enhancer/promoter fragment (*cdh-3>cki-1::SL2::mNG*) partially rescued the AC proliferation and invasion defects caused by *nhr-67* RNAi (**Fig 3E–3G**). By contrast, CKI-1 overexpression did not rescue the invasion defects caused by *egl-43* RNAi (**Fig 3E and 3F**), and it only slightly inhibited AC proliferation (**Fig 3G**). Notably, around 10% of *egl-43* RNAi treated and CKI-1 overexpressing animals contained one AC, yet their BMs were not breached (**Fig 3F**).

These data suggested that that *egl-43* acts in parallel with *nhr-67* and *hda-1* to maintain the G1 arrest and promote invasion of the AC.

## *egl-43* inhibits AC proliferation by repressing *lin-12 Notch* expression

LIN-12 Notch signaling is not only critical during the AC/VU decision, but it also links differentiation to cell cycle progression in different tissues [4,26,27]. We therefore tested whether

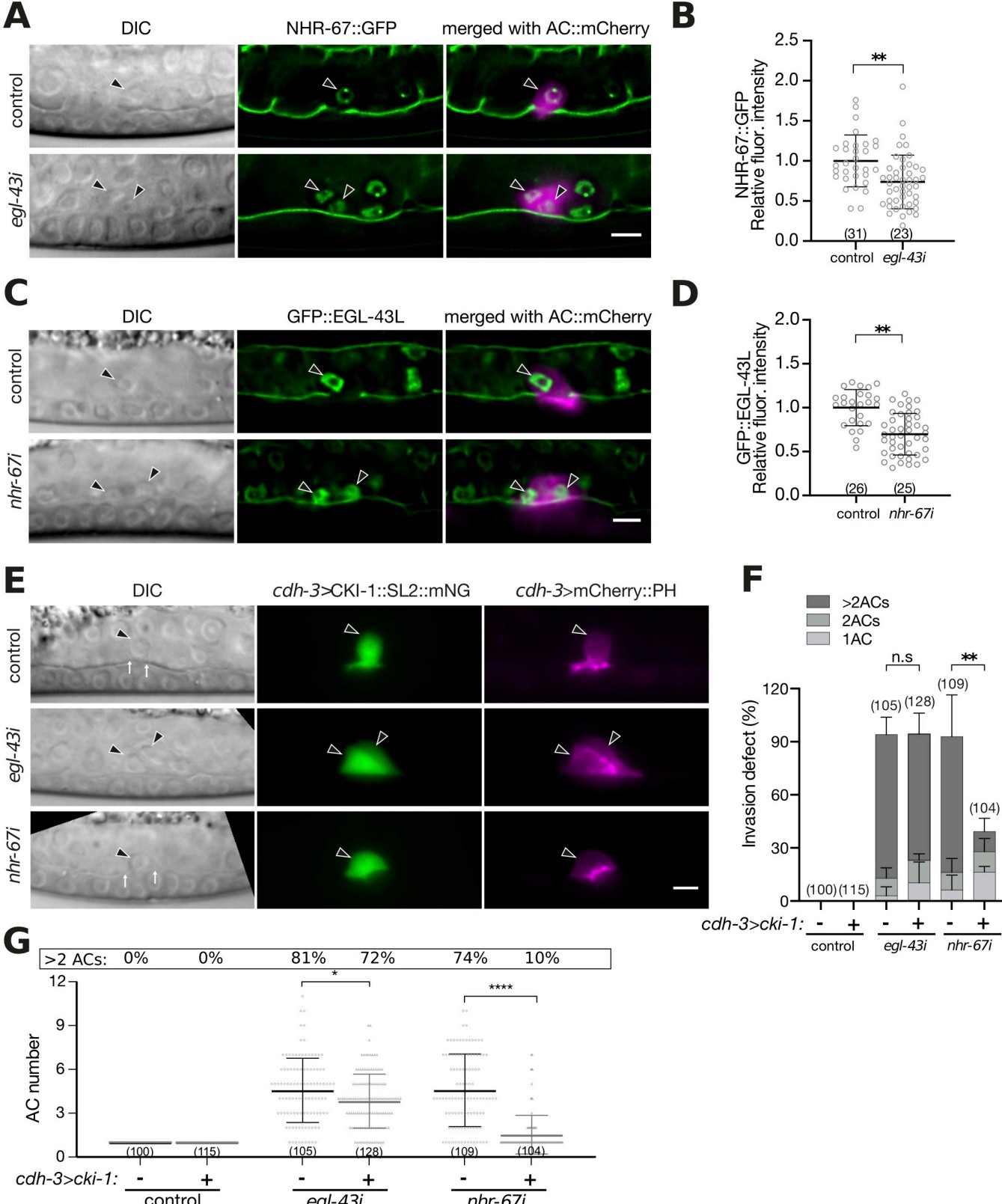

**Fig 3. *egl-43* functions in parallel with *nhr-67* and *cki-1*. (A)** Expression of GFP::EGL-43L after control and *nhr-67* RNAi. The left panels depict Nomarski (DIC) images and the middle panels GFP::EGL-43L expression together with the LAM-1::GFP BM marker at the mid-L3 (Pn.pxx) stage. The right panels show

the GFP signals merged with the ACs labelled by the *lin-3*$^{ACEL}$*>mCherry* reporter in magenta. **(B)** Quantification of GFP::EGL-43L levels in the AC. **(C)** Expression of NHR-67::GFP after control and *egl-43* RNAi. The left panels depict Nomarski (DIC) images and the middle panels NHR-67::GFP expression together with the LAM-1::GFP BM marker at the mid-L3 (Pn.pxx) stage. The right panels show the GFP signals merged with the ACs labelled by the *cdh-3>mCherry::PH* reporter in magenta. **(D)** Quantification of NHR-67::GFP levels in the AC. **(E)** AC-specific expression of *cki-1* from the *cdh-3* enhancer/promoter in control, *egl-43* and *nhr-67* RNAi-treated mid-L3 larvae. The left panels depict Nomarski (DIC) images, the middle panels *cdh-3>*CKI-1::SL2::mNG expression in green and the right panels the ACs labelled by the *cdh-3>*mCherry::PH reporter in magenta. Only animals showing mNG expression in the AC were scored. **(F)** Quantification of the AC invasion and **(G)** proliferation phenotypes in RNAi-treated animals expressing *cdh-3>*CKI-1::SL2::mNG (+) compared to their control siblings lacking the *cdh-3>cki-1*::SL2::mNG transgene (-). The error bars indicate the standard deviation and the horizontal bars the mean values. Statistical significance was determined with a Student's t-test and is indicated with n.s. for p>0.05, * for p<0.05, ** for p<0.01, and **** for p<0.0001. The black arrowheads point at the AC nuclei. The numbers in brackets in the graphs refer to the numbers of animals analyzed. The scale bars are 5 μm.

*egl-43* regulates *lin-12 Notch* expression in the AC by examining a translational LIN-12::GFP reporter [28]. In control animals at the mid-L3 stage, LIN-12::GFP expression had disappeared in the AC, while expression persisted in the adjacent VU cells (**Fig 4A**). By contrast, the multiple ACs that formed after *egl-43* RNAi continued to express LIN-12::GFP (**Fig 4A**). The ACs in *egl-43* RNAi-treated larvae still expressed a *lin-3::mNG* reporter, which serves as a marker to distinguish the AC from the VU fate [29], as well as a reporter for the LIN-12 ligand LAG-2 (**S4A and S4B Fig**) [30]. Thus, the inhibition of *egl-43* did not cause a transformation of the AC into a VU fate, but rather resulted in the ectopic expression of LIN-12 in the proliferating ACs.

To test whether an over-activation of *lin-12* Notch signaling in the AC is responsible for the AC proliferation phenotype, we performed double RNAi of *egl-43* and *lin-12* and scored the number of ACs, as well as their ability to invade. While 63% of *egl-43* single RNAi-treated animals formed multiple (i.e. more than one) ACs, only 16% of *egl-43*; *lin-12* double RNAi-treated animals contained multiple ACs. The average number of ACs per animals decreased from 2.6 in *egl-43* single to 1.2 in *egl-43; lin-12* double RNAi-treated animals (**Fig 4B**), and the penetrance of the AC invasion defect decreased from 72% to 14% (**Fig 4C**). Eight out of the 19 animals that had been treated with *egl-43; lin-12* double RNAi and exhibited an invasion defect contained a single AC. This suggested that the *egl-43* invasion phenotype is not exclusively caused by the over-proliferation of the AC.

To test if reducing *lin-12* activity restored the cell cycle arrest of the AC, we examined the GFP::MCM-7 reporter, which is expressed exclusively in proliferating cells. Control or single *lin-12* RNAi did not induce GFP::MCM-7 expression in the AC of mid-L3 larvae, while *egl-43* RNAi resulted in the formation of multiple GFP::MCM-7 positive ACs in 91% of the cases (**Figs 2A & 4D**). By contrast, the single ACs formed in *egl-43; lin-12* double RNAi-treated animals did not express GFP::MCM-7 in 68% of the cases (**Fig 4D**).

LIN-12::GFP expression was also up-regulated in the AC of *nhr-67* RNAi treated animals (**Fig 4A**). However, inhibition of *lin-12* only partially suppressed the AC over-proliferation caused by *nhr-67* RNAi from 3.8 to 2.6 ACs per animal (**Fig 4B**) and only slightly reduced the *nhr-67* invasion defects from 94% to 81% (**Fig 4C**).

Thus, reducing *lin-12* activity suppressed the AC proliferation and invasion defects caused by *egl-43* RNAi. By contrast, the *nhr-67* phenotype was less sensitive to a reduction in *lin-12* levels, suggesting that NHR-67 inhibits AC proliferation predominantly through another pathway.

## Activation of LIN-12 Notch signaling in the differentiated AC triggers proliferation

These results led to the hypothesis that the ectopic activation of LIN-12 Notch signaling caused by loss of *egl-43* function may trigger the re-entry of the AC into the cell cycle. In order to test

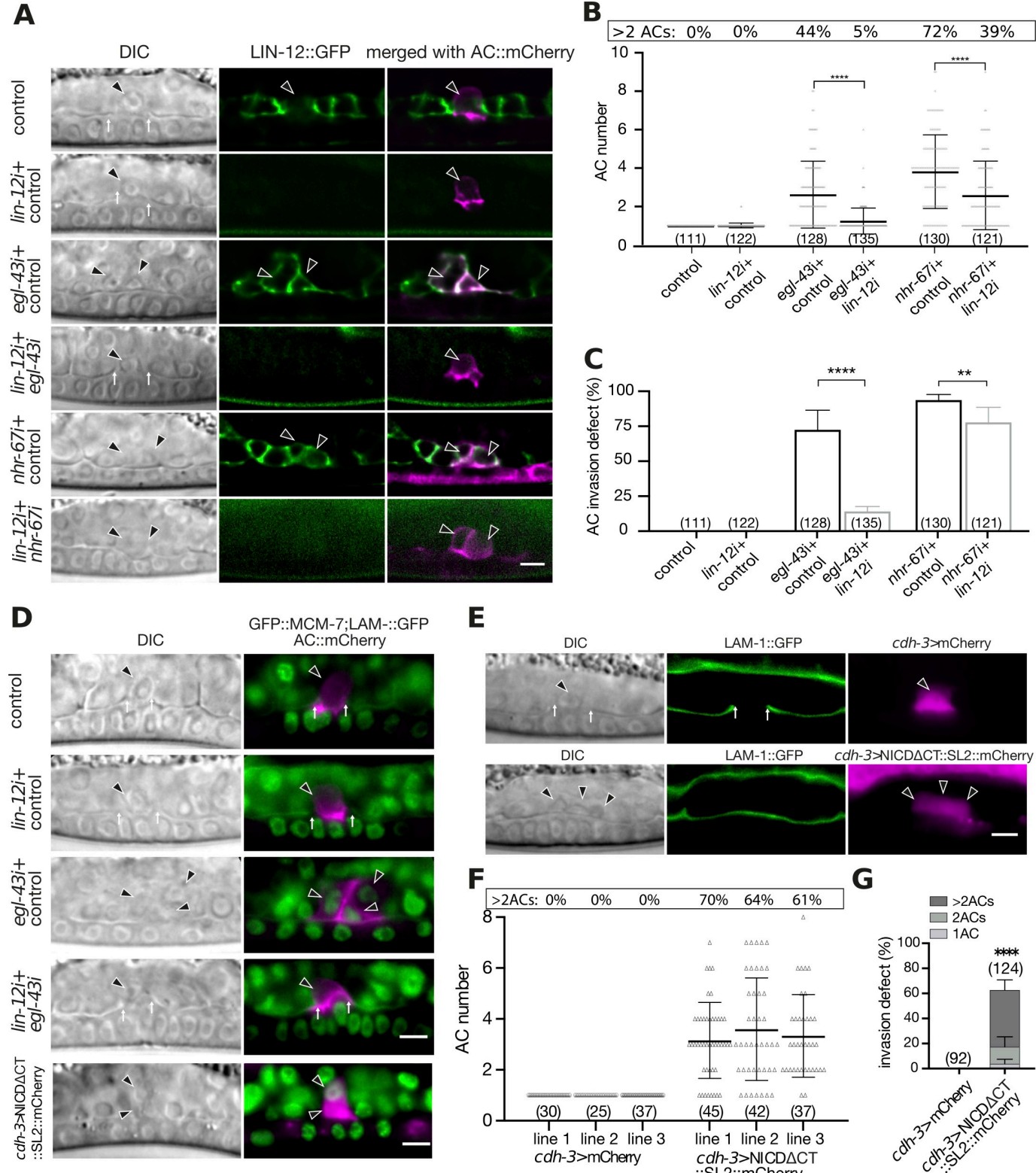

**Fig 4. *egl-43* and *nhr-67* repress *lin-12* Notch expression to prevent AC proliferation. (A)** Expression of LIN-12::GFP after control, *lin-12*, *egl-43* or *nhr-67* single and *egl-43; lin-12* or *nhr-67; lin-12* double RNAi. The left panels shows Nomarski (DIC) images, middle panels LIN-12::GFP expression in green and the right panels the GFP signal merged with the AC labelled by the *cdh-3>mCherry::PH* reporter in magenta. Elevated LIN-12::GFP expression was observed in the ACs of 93/128 *egl-43* and 119/130 *nhr-67* RNAi treated animals, while none of 111 control RNAi animals showed LIN-12::GFP expression in the AC. **(B)**

Quantification of the AC numbers in mid-L3 larvae treated with the different RNAi combinations shown in **(A)**. **(C)** Quantification of the AC invasion defects caused by the different RNAi treatments shown in **(A)**. **(D)** GFP::MCM-7 expression together with LAM-1::GFP after control, *egl-43* or *lin-12* single and *egl-43; lin-12* double RNAi, and in larvae expressing NICDΔCT::mCherry. The left panels show Nomarski (DIC) images and the right panels the GFP::MCM-7 signal in green merged with the AC labelled by the *cdh-3>mCherry::PH* reporter in magenta. 32/35 *egl-43* single RNAi and 13/40 *egl-43; lin-12* double RNAi treated animals showed GFP::MCM-7 expression in the AC. None of the 32 control and of the 40 *lin-12* single RNAi treated animals exhibited GFP::MCM-7 expression in the AC. Expression of NICDΔCT::mCherry induced GFP::MCM-7 expression in 32/33 cases. **(E)** AC-specific expression of *nicdΔct* from the *cdh-3* enhancer/ promoter leads to the formation of multiple ACs. Left panels shows Nomarski (DIC) images and middle panels the BM marker LAM-1::GFP. The right panels show the mCherry expression from the *cdh-3* promoter as a control (row 1) or co-expressed with *nicdΔct* from a bi-cistronic transcript (row 2). **(F)** Quantification of the AC numbers in three independent control lines expressing mCherry alone and in three lines expressing NICDΔCT together with mCherry in the AC. **(G)** Quantification of the AC invasion defects in control lines and in lines expressing NICDΔCT together with mCherry in the AC. The pooled results of the three indepdnet lines are shown. The error bars indicate the standard deviation and the horizontal bars in **(B)** and **(F)** the mean values. Statistical significance was determined with a Student's t-test and is indicated with ** for p<0.01 and **** for p<0.0001. The black arrowheads point at the AC nuclei. The numbers in brackets in the graphs refer to the numbers of animals analyzed. The scale bars are 5 μm.

if LIN-12 signaling is sufficient to induce AC proliferation, we expressed a fragment of the intracellular LIN-12 Notch domain, in which the C-terminal PEST degradation motif had been deleted (NICDΔCT) [26], under the control of the AC-specific *cdh-3* enhancer/promoter fragment. Expression of NICDΔCT in the VPCs was shown to hyper-activate the Notch signaling pathway [26], and *cdh-3*-driven expression occurs only after the AC fate has been specified [31]. Three independent transgenic lines carrying the *cdh-3>nicdΔct::SL2::mCherry* transgene (co-expressing an mCherry marker on a bi-cistronic mRNA) exhibited an AC proliferation phenotype with an average of 3.4 ACs per animal, which is comparable to the phenotype observed after *egl-43* RNAi (**Fig 4E and 4F**). In addition, the GFP::MCM-7 reporter was upregulated (**Fig 4D**), even though GFP::EGL-43L continued to be expressed in the multiple ACs of *cdh-3>nicdΔct::SL2::mCherry* animals (**S4C Fig**). In addition, NICDΔCT expression caused a penetrant AC invasion defect (**Fig 4G**).

Thus, the ectopic activation of the LIN-12 Notch pathway in the differentiated AC was sufficient to trigger cell cycle entry. Taken together, we propose that EGL-43 inhibits LIN-12 Notch expression to maintain the G1 arrest of the invading AC.

## A positive regulatory feedback loop between *egl-43* and *fos-1* activates proinvasive gene expression

It has previously been reported that *fos-1* positively regulates *egl-43* expression in the AC and that *egl-43* is required for the expression of *zmp-1*, *cdh-3* and *him-4* [10,11]. To further characterize the role of *egl-43* in regulating pro-invasive gene expression, we investigated a possible mutual regulation of *fos-1* and *egl-43*. The expression of the endogenous GFP::EGL-43L reporter in the AC was reduced approximately two-fold in homozygous *fos-1(ar105)* mutants compared to heterozygous *fos-1(ar105)/+* control siblings at the mid-L3 stage (**Fig 5A and 5B**). Using CRISPR/Cas9 genome editing, we deleted the 11 bp (TTACTCATCTT) FOS-Responsive Element (FRE) [11] in the promoter region of the endogenous *gfp::egl-43L* reporter strain (*ΔFRE>gfp::egl-43L*, **Fig 1A**). In a heterozygous *fos-1(ar105)/+* background, the expression of the mutant *ΔFRE>gfp::egl-43L* reporter was reduced to a similar extent as the wild-type *gfp::egl-43L* reporter was reduced in a homozygous *fos-1(ar105)* background. Since the levels of the FRE mutant reporter did not further decrease in homozygous *fos-1(ar105)* larvae, FOS-1 appears to control *egl-43* expression mainly through this FRE (**Fig 5A and 5B**). While this experiment confirmed that FOS-1 up-regulates endogenous *egl-43* expression, it also pointed at the existence of additional factors that activate *egl-43* expression in the AC. In particular, the deletion of the FRE in *egl-43* did not cause an obvious defect in AC invasion (all 23 animals scored showed normal AC invasion), suggesting that the reduction in *egl-43* expression after the deletion of the FRE can be compensated by the AC.

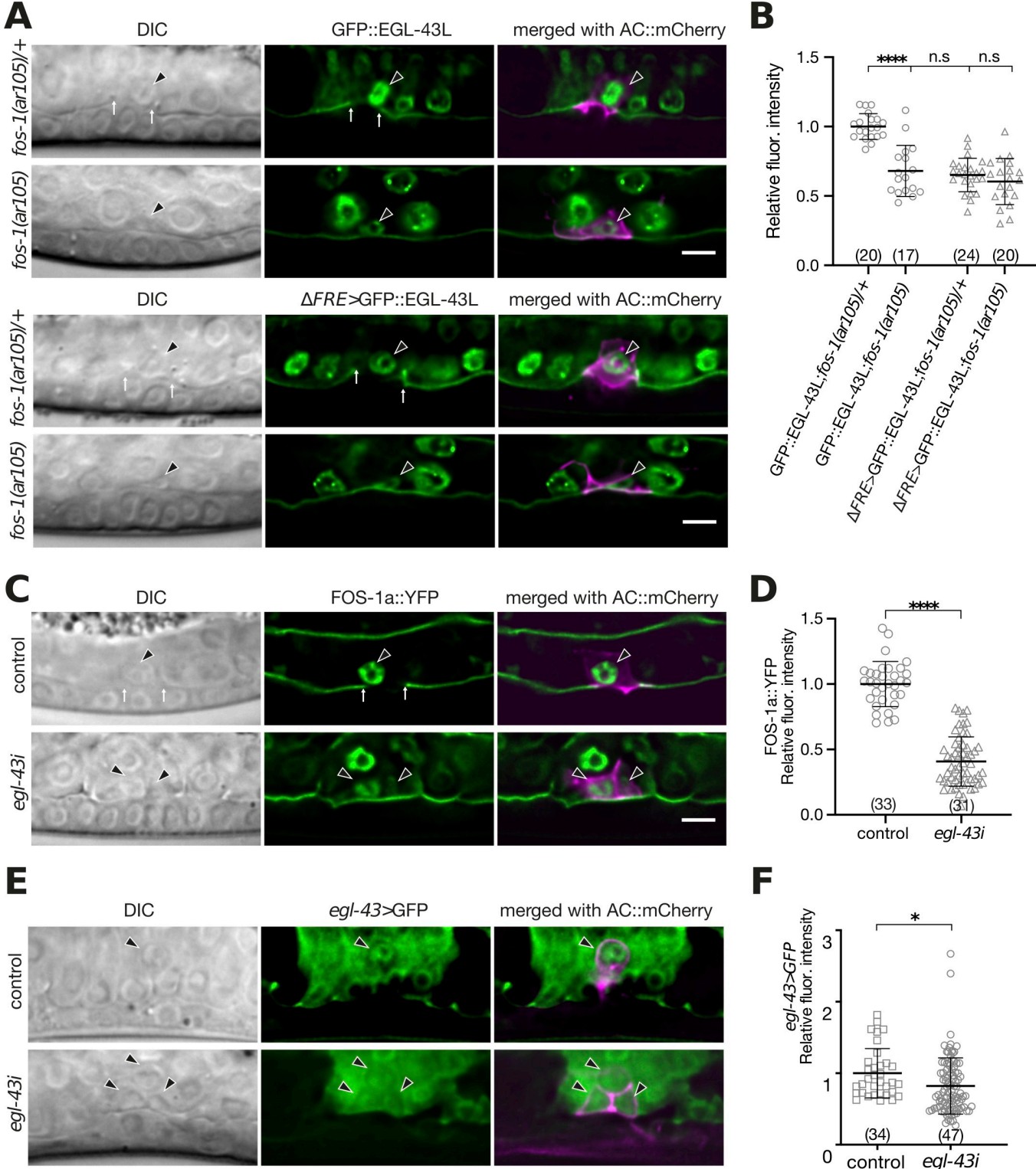

**Fig 5. *egl-43* and *fos-1* regulate each other's expression. (A)** Expression of endogenous GFP::EGL-43L and of the mutant *ΔFRE*>GFP::EGL-43L reporter carrying a deletion of the Fos-responsive element TTACTCATCTT (ΔFRE), each in a *fos-1(ar105)/+* heterozygous and *fos-1(ar105)* homozygous background at the mid-L3 stage. **(B)** Quantification of GFP::EGL-43L expression levels in the ACs of the indicated mutant backgrounds. **(C)** Expression of a FOS-1a::YFP reporter after control and *egl-43* RNAi. **(D)** Quantification of FOS-1a::YFP levels in the ACs after control RNAi. **(E)** Expression of a transcriptional *egl-43>gfp* reporter in the ACs after control and *egl-43* RNAi. **(F)** Quantification of the transcriptional *egl-43>gfp* reporter expression shown in **(E)**. For each reporter, the

left panels show Nomarski (DIC) images, the middle panels the GFP signals of the indicated reporters in green (in (**A**) and (**C**) together with the LAM-1::GFP BM marker) and the right panels the GFP reporter signals merged with the ACs labelled with the *cdh-3>mCherry*::*PH* reporter in magenta. The black arrowheads point at the AC nuclei and the white arrows at the locations of the BM breaches. The error bars indicate the standard deviation and the horizontal bars the mean values. Statistical significance was determined with a Student's t-test and is indicated with n.s. for p>0.05, * for p<0.05 and **** for p<0.0001. The numbers in brackets refer to the numbers of animals analyzed. The scale bars are 5 μm.

Since *egl-43* and *fos-1* regulate some of the same target genes [11,15], we tested if *egl-43* regulates *fos-1a* expression. The expression of a *fos-1a*::*yfp* reporter in the AC was reduced approximately three-fold by *egl-43* RNAi (**Fig 5C and 5D**), indicating that *egl-43* and *fos-1* positively regulate each other's expression. Moreover, the expression of a transcriptional *egl-43L* reporter (*egl-43L>gfp* is a strain containing an insertion of the self-excising cassette [16] after the *gfp* coding sequences to terminate transcription 5' of the *egl-43* coding sequences) was reduced by *egl-43* RNAi (**Fig 5E and 5F**). Thus, *egl-43* positively regulates its own expression in the AC.

Unlike *nhr-67* or *egl-43*, *fos-1* was not required to maintain the G1 arrest of the AC, as neither the S-phase maker RNR-1::GFP nor the proliferation marker GFP::MCM-7 were up-regulated after *fos-1* RNAi (**S5A and S5B Fig**). We thus speculated that the regulation of *fos-1* by EGL-43 and the cell cycle inhibition via *lin-12* repression represent two independent functions of EGL-43. Supporting this hypothesis, LIN-12::GFP expression was not up-regulated by *fos-1* RNAi, while expression of NICDΔCT did not affect *fos-1* expression levels in the AC (**S5C–S5E Fig**).

Taken together, the expression analysis indicated that *egl-43* and *fos-1* form a positive feedback loop that maintains high expression of both transcription factors in the AC to induce the expression of target genes required for invasion. The auto-regulation of *egl-43* likely adds further robustness to this network.

## EGL-43 binding is enriched at the *fos-1*, *egl-43* and *lin-12* loci

To test if the changes in *fos-1*, *lin-12* and *egl-43* reporter expression caused by *egl-43* RNAi could be due to a direct regulation by EGL-43, we performed chromatin immuno-precipitation and sequencing (ChIP-seq) analysis of the endogenous EGL-43::GFP reporter at the L3 stage using anti-GFP antibodies, in two biological replicates (see extended methods). At this stage, most of the EGL-43::GFP expression was confined to cells in the somatic gonad and to approximately 30 head and 6 tail neurons [11] (and this study). The ChIP-seq analysis identified 6276 peaks of significant enrichment, 5257 of which we could associate with 3977 genes (**S5 Table**).

EGL-43 binding was found at the *egl-43*, *fos-1* and *lin-12* loci (**Fig 6**). Notably, EGL-43 was also enriched at the *jun-1* locus, which encodes the homolog of the human c-JUN proto-oncogene that forms together with c-FOS the AP-1 transcription factor (**S5 Table**). Even though no function of JUN-1 in AC invasion has been reported, this observation might indicate that EGL-43 regulates AP-1 activity in other processes, such as ovulation or lifespan [32]. Moreover, EGL-43::GFP was enriched at other previously reported targets, including *mig-10* [33], *hlh-2* [34] and *zmp-1* [11] (**S5 Table**). On the other hand, no specific binding to the *nhr-67* [34] or *cdh-3* [11] genes was observed, suggesting that these two genes may be indirectly regulated by EGL-43.

Taken together, the ChIP-seq analysis suggested that *fos-1*, *lin-12* and *egl-43* are direct EGL-43 targets.

## Discussion

An uncontrolled activation of cell invasion is one of the hallmarks of malignant cancer cells that form metastases [1]. Genetic studies in model organisms have indicated that invasive

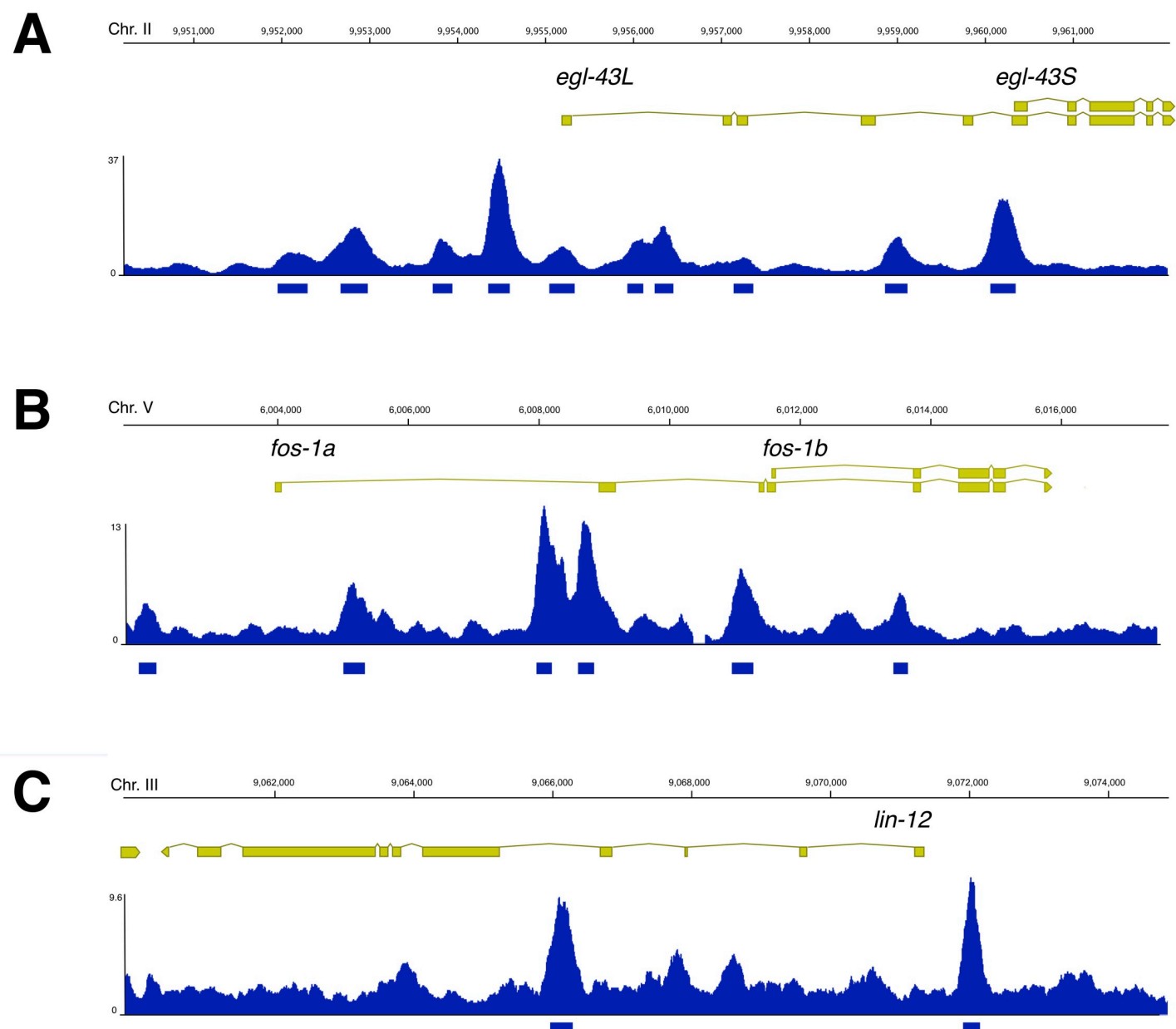

**Fig 6. EGL-43 binding is enriched at the *egl-43*, *fos-1* and *lin-12* loci. (A)** Enrichment of EGL-43 binding at the *egl-43*, **(B)** *fos-1* and **(C)** *lin-12* loci. The exon-intron structure indicated by the yellow boxes and the genomic locations in base pairs of the three analyzed genes are shown on top. The blue bar graphs show the combined data of two independent ChIP-seq experiments. The numbers on the left vertical axis of each graph indicate the maximal read coverage in the intervals shown. The blue shaded boxes underneath the graphs show the identified peaks. S5 Table contains a list of all identified EGL-43 binding sites.

tumor cells re-activate the same molecular pathways that control cell invasion during normal animal development. AC invasion in *C. elegans* has served as an excellent model to dissect the genetic pathways regulating the invasive phenotype of a single cell [3].

Here, we report that the *egl-43* gene, the *C. elegans* ortholog of the human *Evi1* proto-oncogene, functions in a regulatory network together with the transcription factors NHR-67 and FOS-1 to control AC invasion (**Fig 7**). Our results are consistent with a recent report by Medwig-Kinney et al. [34] demonstrating similar interactions between *egl-43*, *nhr-67* and *fos-1* during AC invasion. The inclusion of the LIN-12 NOTCH pathway in our model further expands

the network and differentiates between the functions of EGL-43 and the nuclear receptor NHR-67. Even though *egl-43* and *nhr-67* positively regulate each other's expression at a later stage, our data indicate that EGL-43 and NHR-67 inhibit AC proliferation through distinct mechanisms.

Firstly, EGL-43 maintains the AC arrested in the G1 phase of the cell cycle by repressing the expression of the LIN-12 Notch receptor. The ectopic activation of Notch signaling in the differentiated AC was sufficient to induce proliferation in the presence of EGL-43, while reducing *lin-12* expression efficiently suppressed the AC proliferation and invasion defects caused by inhibition of *egl-43*. Thus, LIN-12 is an essential downstream target of EGL-43 that can reactivate the cell cycle in the AC. A recent study in *C. elegans* has highlighted the importance of the LIN-12 Notch pathway in keeping an equilibrium between the proliferation and differentiation of somatic cells [27]. Furthermore, the regulation of different cell cycle genes by the Notch pathway has been reported in several cases. For example, Notch signaling induces cyclin D1 expression in mammalian kidney and breast epithelial cells [35,36] and activates dE2F1 and cyclin A expression in the photoreceptor precursors of the *Drosophila* eye to promote S-phase entry [37].

It was previously shown that NHR-67 maintains the G1 arrest of the AC by inducing the expression of the CDK inhibitor CKI-1 [8]. While overexpression of CKI-1 efficiently rescued the AC proliferation and invasion phenotype caused by *nhr-67* RNAi, CKI-1 only slightly suppressed the AC proliferation phenotype and had no effect on the invasion defects caused by *egl-43* RNAi. However, it should be noted that the transgene we used to overexpress CKI-1 did not fully suppress the *nhr-67* AC proliferation and invasion phenotypes, while a different *cki-1* overexpression transgene used by Medwig-Kinney et al. [34] caused a complete rescue of *nhr-67* RNAi, probably due to higher levels of CKI-1 expression. It is therefore possible that a further increase in CKI-1 levels beyond the concentration we reached may also suppress the *egl-43* phenotype. Taken together, we suggest that the AC proliferation phenotype caused by inhibition of *egl-43* is less sensitive to an increase in the CKI-1 dosage than the *nhr-67* phenotype.

On the other hand, the *nhr-67* AC proliferation and invasion phenotypes were less sensitive to *lin-12* inhibition when compared to the *egl-43* phenotype, even though NHR-67 RNAi also increased *lin-12* expression in the AC. We thus propose that EGL-43 inhibits AC proliferation predominately by repressing LIN-12 NOTCH signaling, while NHR-67 acts primarily by enhancing CKI-1 expression (**Fig 7**). One possible explanation for the different sensitivities could be that the hyper-activation of LIN-12 signaling caused by loss of *egl-43* results in elevated CDK/Cyclin activity, which overcomes a threshold set by CKI-1-mediated cell cycle inhibition. The inhibition of *nhr-67*, on the other hand, may reduce the threshold by decreasing CKI-1 expression. Since *nhr-67* and *egl-43* positively regulate each other's expression in the proliferating ACs, EGL-43 may promote *cki-1* expression indirectly via *nhr-67*.

The *egl-43* AC invasion and proliferation phenotypes did not completely correlate, as a fraction of *egl-43* depleted animals -especially in combination with *lin-12* RNAi- contained a single AC that failed to invade. Hence, EGL-43 appears to have an additional function besides merely preventing AC proliferation. EGL-43 likely exerts the proliferation-independent function through its interaction with *fos-1*, which is not required for the G1 arrest of the AC but necessary to induce the expression of pro-invasive genes. Deleting the FOS-1 responsive element (FRE) in the endogenous *egl-43* locus confirmed our earlier findings based on transgenic reporter analysis that *egl-43L* expression is positively regulated by FOS-1 [11]. FOS-1 is not absolutely required for the expression of EGL-43, because additional factors such as HLH-2 activate *egl-43* expression in the AC [10]. Moreover, EGL-43 positively regulates *fos-1* as well as its own expression in the AC. Thanks to the positive feedback loop between *egl-43* and *fos-1*, low levels of either of the two transcription factors may be sufficient to induce a stable

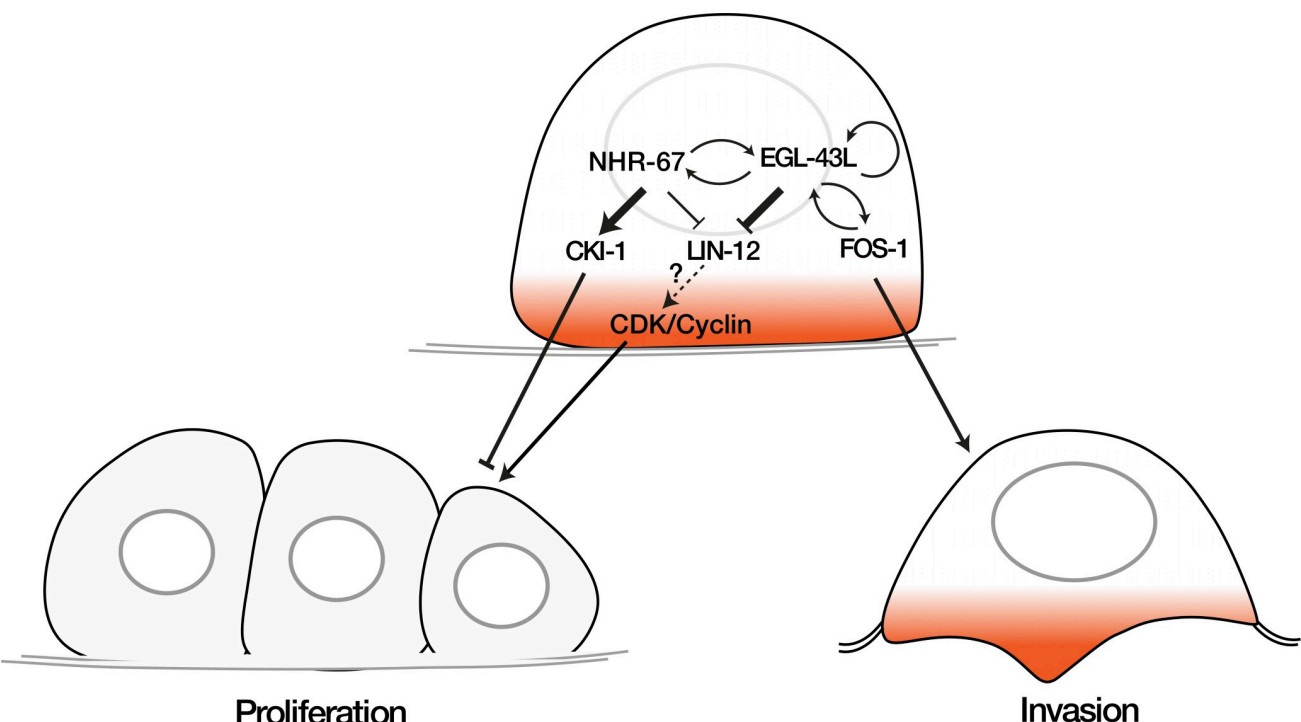

**Fig 7. EGL-43L is part of a regulatory network controlling AC invasion.** EGL-43L plays two distinct functions during AC invasion. Left side: EGL-43L represses LIN-12 Notch expression in the differentiated AC to prevent proliferation. In addition, EGL-43 and NHR-67 positively regulate each other, and NHR-67 controls CKI-1 expression to maintain the G1 arrest of the AC. Right side: EGL-43L activates in a positive feedback loop together with FOS-1 the expression of pro-invasive genes in the AC.

expression of both transcription factors and thereby irreversibly determine the invasive fate of the AC. Interestingly, a similar self-activating function has been described for mammalian *Evi1* in hematopoietic stem cells [12].

In summary, EGL-43 coordinates the expression of pro-invasive genes with the cell cycle arrest of the AC by inducing *fos-1* and inhibiting *lin-12 Notch* expression. Thus, EGL-43 is a central component in a regulatory network, which decides whether to divide or invade.

## Material and methods

### *C. elegans* culture and maintenance

*C. elegans* strains were maintained at 20˚C on standard nematode growth plates as described [38]. The wild-type strain was *C. elegans* Bristol, variety N2. We refer to translational protein fusions with a :: symbol between the gene name and the tag, while transcriptional fusions are indicated with a > symbol between the promoter/enhancer and the tag. The genotypes of the strains used in this study are listed in **S1 Table**. The construction of the plasmids, oligonucleotides and sgRNAs used to generate the different reporters are described in the **Extended Methods in S1 Text** and in **S2–S4 Tables**.

### Scoring the AC invasion phenotype

AC invasion was scored in mid-L3 larvae after the VPC had divided twice (Pn.pxx stage) as described [11]. We monitored the continuity of the BM by DIC or fluorescence microscopy using the *qyIs10[lam-1>lam-1::gfp]* transgene as a marker.

## RNA interference

RNAi interference was done by feeding dsRNA-producing *E. coli* [39]. Larvae were synchronized at the L1 stage by hypochlorite treatment of gravid adults and plated on NGM plates containing 3mM IPTG. P0 animals were analyzed after 30–36 hours of treatment. For double RNAi experiments, bacteria of the indicated clones were mixed at a 1:1 ration. RNAi clones targeting genes of interest were obtained from the *C. elegans* genome-wide RNAi library or the *C. elegans* open reading frame (ORFeome) RNAi library (both from Source BioScience). RNAi vectors targeting *egl-43L* and *egl-43S* were subcloned into the L4440 vector by Gibson assembly (**S3 Table**). The empty L4440 vector (labelled "control" in the figures) was used as negative control in all experiments.

## Microscopy and image analysis

Fluorescent and Nomarski images were acquired with a LEICA DM6000B microscope equipped with a Leica DFC360 FX camera and a 63x (N.A. 1.32) oil-immersion lens, or with an Olympus BX61 wide-field microscope equipped with a X-light spinning disc confocal system, a 100x Plan Apo (N.A. 1.4) lens, a lumencor light engine as light source and an iXon ultra888 EMCCD camera. Worms were imaged with 100 x magnification and z-stacks with a spacing of 0.1 to 0.8 μm were recorded. The Fiji software [40] was used for image analysis and fluorescent intensity quantifications using the built-in measurement tools as follows. To quantify expression of the different reporters, deconvolved optical z-sections across the AC were used to generate summed z- projections. The region of the AC nucleus was manually selected, and the integrated signal intensity was measured in the AC nucleus, from which a background value measured in an identically sized region outside of the animal was subtracted. To quantify the CDK sensor activity, images were processed with a Gaussian blur filter (sigma = 50), and a single mid-sagittal z-slice through the AC nucleus was selected for the measurements. The average of the integrated intensities in three equally sized and randomly selected areas, each in the cytoplasm ($I_c$) and the nucleus ($I_n$) of the AC were measured to calculate the cytoplasmic to nuclear ($I_c/I_n$) intensity ratios, which are plotted in **Fig 2F**.

## Supporting information

**S1 Fig. Structure of the FRT-tagged *gfp::egl-43L* allele *zh144*.**
(TIF)

**S2 Fig. The PR domain deletion reduces *egl-43* expression levels. (A)** Quantification of GFP::EGL-43L and GFP::EGL-43LΔPR and **(B)** EGL-43LΔS::GFP and EGL-43LΔZF1::GFP expression levels in the AC of mid-L3 larvae. N- and C-terminal GFP fusions were quantified separately because the site of insertion potentially affects GFP signal intensity. **(C)** Quantification of the AC expression levels of EGL-43::GFP, a reporter for both the EGL-43L and EGL-43S isoforms, upon control, *egl-43*, *egl-43L*, and *egl-43S* RNAi. The numbers of animals analyzed for each condition are shown in bracket. The error bars indicate standard deviations and the horizontal bars the mean values. Statistical significance was determined by Student's t-tests and is indicated with n.s. for p>0.05 and **** for p<0.0001.
(TIF)

**S3 Fig. Regulation of *egl-43*, *nhr-67* and *hda-1* expression in early L3 larvae. (A)** NHR-67::GFP expression in the ACs of early-L3 larvae (Pn.p stage) after *egl-43* RNAi. **(B)** Quantification of NHR-67::GFP expression levels after *egl-43* RNAi. **(C)** GFP::EGL-43L expression in the ACs of early-L3 larvae after *nhr-67* RNAi. **(D)** Quantification of GFP::EGL-43 expression levels after *nhr-67* RNAi. **(E)** HDA-1::RFP expression in mid-L3 larvae after *egl-43* RNAi. **(F)**

Quantification of the HDA-1::RFP expression shown in **(E)**. For all reporters, left panels show Nomarski (DIC) images, middle panels the respective reporter together with the LAM-1::GFP BM marker, and right panels merged images with the ACs labelled by *cdh-3>mCherry*::*PH* **(A)**, *cdh-3>mCherry*::*moeABD* **(C)** or *cdh-3>gfp* **(E)**. The error bars indicate standard deviations and the horizontal bars the mean values. Statistical significance was determined with a Student's t-test and is indicated with ** for p<0.01 and n.s. for p>0.05. The numbers in brackets refer to the numbers of animals analyzed. The scale bars are 5 μm.
(TIF)

**S4 Fig. AC fate markers remain expressed after *egl-43* RNAi, while Notch signaling does not affect EGL-43L expression. (A)** *lin-3* reporter expression in the control and *egl-43* RNAi ACs. Left panels show Nomarski (DIC) images and right panels the fluorescence image with LIN-3::mNG. **(B)** *lag-2* expression in the ACs of control and *egl-43* RNAi. Left panels shows Nomarski (DIC) images, middle panel the fluorescence image with *lag-2>*GFP reporter, and right panels the reporter merged with AC marker *cdh-3>*mCherry::PH. **(C)** GFP::EGL-43L expression in the ACs of control and NICDΔCT expressing ACs. Left panels show Nomarski (DIC) images, middle panels the GFP::EGL-43 signal with the LAM-1::GFP BM marker, and right panels merged with the ACs labelled with *cdh-3>*PH::mCherry (control, row 1) and *cdh-3>*NICDΔCT::SL2::mCherry (row 2) respectively. **(D)** Quantification of the GFP::EGL-43L expression shown in **(C)**. The error bars indicate standard deviations and the horizontal bars the mean values. Statistical significance was determined with a Student's t-test and is indicated with n.s. for p>0.05. The numbers in brackets refer to the numbers of animals analyzed. The scale bars are 5 μm.
(TIF)

**S5 Fig. FOS-1 neither regulates cell cycle markers nor LIN-12 expression, while LIN-12 does not regulate FOS-1 expression. (A)** Expression of the S-phase marker RNR-1::GFP after control and *fos-1* RNAi. None of 19 control or 23 *fos-1i* animals showed RNR-1::GFP expression in the AC. **(B)** Expression of GFP::MCM-7 after control and *fos*-RNAi. None of 25 control or 25 *fos-1i* animals showed GFP::MCM-7 expression. **(C)** LIN-12::GFP expression is not up-regulated after control (0/20) or *fos-1* (0/24) RNAi treatment. **(D)** Expression of FOS-1a::YFP in control and NICDΔCT-expressing ACs of mid-L3 larvae. **(E)** Quantification of the FOS-1a::YFP expression shown in **(D)**. For each reporter, the left panels show Nomarski (DIC) images, the middle panels the GFP or YFP signals of the indicated reporters in green (in **(B)** and **(D)** together with the LAM-1::GFP BM marker) and the right panels the GFP reporter signals merged with the ACs labelled with the *cdh-3>mCherry*::*moeABD* **(A, C)**, *lin-3^ACEL^>mCherry* **(B)** or *cdh-3>nicdΔct*::*sl2*::*mCherry* **(D)** reporters in magenta. The black arrowheads point at the AC nuclei and the white arrows at the locations of the BM breaches. The error bars indicate standard deviations and the horizontal bars the mean values. Statistical significance was determined with a Student's t-test and is indicated with n.s for p>0.05. The numbers in brackets refer to the numbers of animals analyzed. The scale bars are 5 μm.
(TIF)

**S1 Table. List of strains used.**
(DOCX)

**S2 Table. Design of plasmids used.**
(DOCX)

**S3 Table. Oligonucleotide primers used.**
(DOCX)

**S4 Table. Sequences of the guide RNAs used.**
(DOCX)

**S5 Table. List of EGL-43 binding sites identified by ChIP-seq analysis.**
(XLSX)

**S1 Text. Extended methods.**
(DOCX)

## Acknowledgments

We wish to thank members of the Hajnal laboratory for numerous discussions, the *Caenorhabditis Genetics Center*, which is funded by NIH Office of Research Infrastructure Programs (P40 OD010440), and the van der Heuvel laboratory for providing strains. We are also grateful to Andrew Fire for making *gfp* vectors available.

## Author Contributions

**Conceptualization:** Ting Deng, Alex Hajnal, Evelyn Lattmann.

**Data curation:** Ting Deng, Przemyslaw Stempor.

**Formal analysis:** Ting Deng, Przemyslaw Stempor, Evelyn Lattmann.

**Funding acquisition:** Julie Ahringer, Alex Hajnal.

**Investigation:** Ting Deng, Przemyslaw Stempor, Alex Appert, Michael Daube, Evelyn Lattmann.

**Methodology:** Ting Deng, Przemyslaw Stempor, Alex Appert, Michael Daube, Evelyn Lattmann.

**Project administration:** Alex Hajnal.

**Supervision:** Julie Ahringer, Alex Hajnal, Evelyn Lattmann.

**Visualization:** Ting Deng, Evelyn Lattmann.

**Writing – original draft:** Ting Deng, Evelyn Lattmann.

**Writing – review & editing:** Julie Ahringer, Alex Hajnal.

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
