## [Decision Letter · Decision Letter 0]

6 Nov 2019

Dear Dr Hajnal,

Thank you very much for submitting your Research Article entitled 'The C. elegans homolog of the Evi1 proto-oncogene, egl-43, coordinates G1 cell cycle arrest with pro-invasive gene expression during anchor cell invasion' to PLOS Genetics. Your manuscript was fully evaluated at the editorial level and by three independent peer reviewers. The reviewers appreciated the attention to an important problem, were excited about the potential of the work, but raised some substantial concerns about the current manuscript. Based on the reviews, we will not be able to accept this version of the manuscript, but we would be willing to review a much-revised version.  We cannot, of course, promise publication at that time.

Should you decide to revise the manuscript for further consideration here, your revisions should address the shared concerns of reviewer #2 and #3 on rigor--better quantification and outlining methodology more clearly.  All the points raised by reviewer #1 and #2 should be addressed.  For Reviewer #3, please address points 1 and 3 (it is not clear that 4 can be addressed and 5 and 6 are beyond the scope of the work). We will require a detailed list of your responses to the review comments and a description of the changes you have made in the manuscript.

[GPC Note: two of the reviewers indicated that the data underlying the figures in the manuscript have not been provided.  In particular they mentioned numerical data for the bar graphs, sequence of the egl-43Si 5'UTR used in the generation of the RNAi targeting clone to the short isoform of egl-43, and data on expression of the different GFP-tagged alleles.  PLOS Genetics, as part of its Open Data policy, requires that all such data be provided in the manuscript (as supplemental material), or be deposited in a public database.  Please address this issue in your revised manuscript, and describe the amendments you make in your response-to-review letter.]

If you decide to revise the manuscript for further consideration at PLOS Genetics, please aim to resubmit within the next 60 days, unless it will take extra time to address the concerns of the reviewers, in which case we would appreciate an expected resubmission date by email to plosgenetics@plos.org.

[LINK]

We are sorry that we cannot be more positive about your manuscript at this stage. Please do not hesitate to contact us if you have any concerns or questions.

Yours sincerely,

David R Sherwood

Guest Editor

PLOS Genetics

Gregory P. Copenhaver

Editor-in-Chief

PLOS Genetics

Reviewer's Responses to Questions

**Comments to the Authors:**

Reviewer #1: This article by Deng and coworkers is a study of EGL-43, the C. elegans ortholog of the human Evi1 proto-oncogene, and how it regulates the fate and function of the invading anchor cell (AC). The authors show that egl-43 inhibits AC proliferation by repressing lin-12 expression. (G1 arrest is known to be essential for the AC to realize its invasive potential.) Furthermore the authors show that a positive feed-back loop exists between egl-43 and fos-1 in the AC, maintaining high expression of both for the induction of genes required for AC invasion.

I have no major criticisms of this work. It appears solid and complete, and the text is well-written and clear. The only thing I felt was lacking was more discussion of the role of egl-43 in the fate decision of the AC as compared to other studies on this subject, in particular the recent work from the Greenwald lab (HLH-2/E2A Expression Links Stochastic and Deterministic Elements of a Cell Fate Decision during C. elegans Gonadogenesis, Current Biology, Sept 2019). How does egl-43 function fit in with the HLH-2 expression clock described in this paper?

Reviewer #2: Summary:

The research described in this manuscript examined the function of egl-43 in regulating C. elegans anchor cell (AC) invasion. Using CRISPR/Cas9 mediated genome editing, the authors have engineered several new egl-43 alleles that provide valuable insight into egl-43 function. Specifically, they identify the long isoform (egl-43L) as the predominant isoform functioning during AC invasion as well as regulation of egl-43 by fos-1 via a cis-regulatory FOS-Responsive Element. Using multiple cell cycle reporters, the authors also show that egl-43 is required for the AC to undergo G1 cell cycle arrest, and that egl-43 depletion results in proliferative ACs. The authors claim that this cell cycle dependent function of egl-43 is independent of the nhr-67 / CKI-1 pathway known to be involved in AC invasion, although I feel that this is not well-supported by the evidence currently provided in the manuscript. Opportunities to include more convincing controls and to quantify data were sometimes missed by the authors, though should be easily remedied by either better quantification of existing data or new experiments. Suggested revisions to the manuscript include validation of RNAi reagents, explicit indication of sample sizes, and additional experiments (outlined below). Despite the fact that some of this data (i.e., the relationship between egl-43 and fos-1; egl-43’s role in cell cycle arrest) has been recently shown in a bioRxviv preprint (Medwig-Kinney et al. (2019)), I feel that the mechanistic insights gained through the authors’ careful dissection of the egl-43 locus is complementary and I am enthusiastic about seeing this work published. Furthermore, the novel finding that ectopic expression of Notch intracellular domain is sufficient to induce proliferation in a normally post-mitotic differentiated cell is a very exciting finding and would be of interest to a broad readership.

Comments:

General comments:

Assuming PLoS Genetics allows for citations of preprints, given the nature of overlap between this work and that of a recently updated bioRxviv preprint (Medwig-Kinney et al. (2019)), I think it would be useful for the field if the authors discussed their results in the context of data showing that egl-43 regulates both hlh-2 and nhr-67 in a cell-cycle dependent manner as well as feedback between EGL-43 and FOS-1. The main discrepancy between this work and Medwig-Kinney et al. (2019) is whether or not egl-43 and nhr-67 function independently of each other in mediating G1/G0 arrest in the AC. See below for specific experimental suggestions that might clear this discrepancy up.

I would recommend for showing single channel fluorescence images to use grey scale, which the human eye can see subtle differences in easier than false colored images, and only use false colored images for overlays.

In the results and brief Discussion section the authors miss the chance to put their data using endogenously-tagged alleles in the context of what has been shown by their and other labs previously using transgenes - for example, autoregulation of egl-43 has been shown multiple times based on transgenes and the first potential explanation for this is that levels of egl-43 are extremely important - see Wang et al. 2014 (doi: 10.1016/j.bbrc.2014.08.049) - where they show that egl-43 functions through an incoherent feed forward circuit with negative feedback in regulating MIG-10 levels in the AC.

For the most part, the authors represent fluorescence quantification data through box plots, which depict median values. However, given the wide spread of some of this data (e.g., Fig. 4B), median may not be the best statistic to show. I would recommend using an alternative method of data visualization, such as violin plots including mean values and standard deviation.

Introduction:

Potential typos (minor):

“selected” → “select” (paragraph 1)

“trackable” → “tractable” (paragraph 1)

“EGl-43” → “EGL-43” (paragraph 3)

“the VU cell undergoes three rounds of cell divisions” - This is not 100% accurate, as the ρ cells undergo an extra round of division. See Newman, White, & Sternberg (1996).

Results:

“FRT” should be defined upon 1st use of the acronym. The FRT experiments are really elegant - I’m wondering, is the reduced penetrance in these lines as compared to RNAi due to produrance of the protein during the length of time it takes for the flipase to remove the genomic region flanked by FRT sites? I couldn’t tell from the images - it looks like there is no expression, but it would be useful to quantify this.

“We found no obvious difference in the expression pattern of the two egl-43 reporters, suggesting that the long egl-43L isoform accounts for most of the expression observed.” - This claim can be supported by evidence showing quantitative comparison of expression levels in both reporters (not directly provided).

How was the egl-43Si RNAi construct validated? The targeting sequence is presumably much smaller (although this information is missing from the supplement) than typical RNAi constructs, so the efficiency may be significantly lower. Also, how did the authors determine the 5’ UTR sequence, as I could not find it annotated on WormBase? The authors also may want to consider that there is evidence (Bosher et al., 1999) that RNAi can act on pre-mRNA, which would indicate that this construct may recognize the introns of pre-spliced egl-43L transcripts.

Figure 1B-C: It would be helpful to see quantification of this data presented as well.

How was the sample size for the egl-43L RNAi vs. egl-43 RNAi experiment (Results paragraph #2) determined? Typically a minimum sample size of 28 is required to perform a significance test at ɑ = 0.05.

Figure 2: The number of animals observed with the representative phenotype shown, with respect to the total number of animals observed, should be indicated in Figures 2A,C-D. The n indicated in the bar graph in panel B is difficult to read due to the small font size (and I expect the font size would need to be increased for publication per journal standards anyway).

The source of the RNR-1::GFP strain/construct (Park & Krause, 1999) should be cited in addition to the WormBook chapter.

Is the characterization of the endogenously-tagged MCM-7::GFP described elsewhere? I know that the transgene has been used as a reporter for actively cycling cells by the van den Heuvel lab (I would recommend citing the data paper, Korzelius et al. 2011, rather than the wormbook chapter here). If this is the first description of the endogenous MCM complex as a reporter for S-phase onset/cycling cells it would be worth characterizing it first and then using it as a reporter. I believe the data, I just think it would be nice to highlight that it’s a GFP-knock in - you could cross the allele into the MCM-4::mCherry transgene from the van den Heuvel lab and just demonstrate that they show the same exact pattern of localization in a cell cycle-dependent way as a supplemental figure?

The original CDK biosensor citation should be included as well from Spencer et al. 2011 when citing its use as it was co-opted from mammalian cell culture.

Image quantification: The Materials and Methods section is specifically lacking a description of how the CDK sensor was quantified, and in general more information is needed in reference to image quantification for all of the data in the manuscript - “built-in measurement tools” in Fiji/ImageJ could mean many different things - how did the authors correct for background/camera noise? Were measurements made from single confocal z-planes? Are the authors’ reported mean grey values or integrated density (either is fine, just more details are needed). Did the authors use thresholding and the wand tool to select the region of interest, or did they hand draw regions of interest?

In the updated Medwig-Kinney et al. pre-print, it is shown that the regulatory relationships between egl-43 and nhr-67 do not exist until post-AC-specification. I recognize that this data was not available at the time of submission. However, this could explain why the authors do not see a significant change in nhr-67::GFP expression in the AC following egl-43(RNAi). More importantly, however, I would suggest that the authors examine mitotic ACs for regulation of gene expression rather than looking at earlier stages, as it is impossible to know whether or not an AC is out of cycle (beyond using a second set of reporters for cell cycle state) so you can not assess whether the single AC you are measuring is going to invade or not. As to the authors’ statement that the proliferation of the AC results in dilution of protein expression - this data exists. I would point the authors to data using the transgene containing the full (~5kb) cdh-3 promoter fused to GFP. In Matus et al. (2015), I found that this promoter was expressed at ~97% of wild-type levels in proliferating ACs (see Figure 4B from Matus et al. 2015), while other reporters are clearly down-regulated, suggesting that GFP is not simply diluted as ACs become mitotic and proliferate but that the actual transcriptional program is changing due to inappropriate cell cycle entry.

This also brings up an important point on the use of the cdh-3 promoter for driving constructs of interest in the AC. We have found that the smaller ACEL used to drive cdh-3 (~1.5kb) for many of the transgenes from the Sherwood lab, also used in this paper (qyIs23 and qyIs50), is regulated by nhr-67 and egl-43, which is why we used the full cdh-3 promoter (~5kb) to generate new AC reporters for studying nhr-67 loss of function in Matus et al. 2015. I bring this up because if anything, the use of the smaller cdh-3>constructs could under-report the number of ACs due to depletion of the promoter. It took a little digging for me to figure out that the new constructs were designed with the full cdh-3 promoter, so it would be helpful to distinguish this in the text/methods. We used cdh-31.5 vs cdh-3 in our original paper if that nomenclature is helpful.

Figure 3G,I: The overexpression of CKI-1 in a lineage should cause G1/G0 arrest. It would appear that the authors are making the claim that egl-43 mediates cell cycle arrest independent of CKI-1, but a more parsimonious explanation would be that depletion of egl-43 results in downregulation of the cdh-3 promoter driving CKI-1 expression, and in cases where you see multiple ACs, those ACs do not have a critical threshold of CKI-1 activity to prevent cell cycle entry. One suggestion would be to quantify levels of CKI-1 in all of the animals and see if there is a statistical correlation between CKI-1 levels and number of ACs observed. While, the 2 AC phenotype could be the result of perturbing AC/VU specification, 3+ ACs shouldn’t be observed if the cdh-3>CKI-1 is functioning 100% of the time.

Figure 3G-I: The key says “control siblings” - does this mean that these are the progeny resulting from a cross? I assume not, but this terminology may be misleading and whether animals of homozygous or heterozygous is an important distinction.

Figure 4A: This data should be quantified. Also how was expression of lin-12 determined? How were the boundaries of ACs versus VUs determined in adjacent cells? The endogenously- tagged lin-12::GFP reporter from Attner et al. (2019) has both membrane bound and nuclear localization, making it easier to distinguish which cell has active Notch signaling - this strain might be easier to use and would better make this really stunning point that active Notch signaling post-AC/VU decision can force the differentiated AC into the cell cycle and inhibit invasion.

Figures 4A-C: When was lin-12 RNAi treatment administered? Knockdown of lin-12 prior to AC/VU specification may confound the number of ACs observed. It would be worthwhile to try an L2 plating of lin-12(RNAi) and see if, at some penetrance, you can repeat your experimental results.

Figure 4D: What percentage of animals are the phenotypes shown indicative of?

Figure 5B: It would be helpful to show GFP::EGL-43L expression without fos-1(ar105) in the background here.

The authors postulate that egl-43 has cell cycle dependent (lin-12) and independent (fos-1) roles, but this is not supported by the data showing that lin-12(RNAi) rescues AC invasion in egl-43(RNAi) animals.

The authors should mention the potential effects that the endogenous transcriptional reporters (with pre-floxed SEC) have on protein function.

The introduction and discussion need elaboration, specifically with regard to links between Notch signaling and cell cycle regulation, in C. elegans and other model systems.

The authors argue that egl-43 and nhr-67 control cell cycle arrest in distinct pathways. To show this convincingly, they should perform the lin-12/Notch experiments with nhr-67 RNAi perturbation experiments, the expectation would be that nhr-67(RNAi) does not induce lin-12 expression if the two pathways are independent. Alternatively, as we believe that egl-43 does regulate nhr-67 activity, it would be interesting if this was still the case - that nhr-67(RNAi) does not regulate lin-12, as we have recently shown that endogenous lin-12::GFP is strongly down-regulated pre-AC/VU decision in our bioRxiv preprint. If you find that nhr-67(RNAi) doesn’t turn on lin-12::GFP, it could also suggest that egl-43 has nhr-67-dependent and nhr-67-independent roles in maintaining the AC in a post-mitotic state, and provide an explanation why nhr-67(RNAi) on an nhr-67(pf88) hypomorphic allele doesn’t significantly increase the AC invasion/proliferation defect (it makes it slightly worse, but there are still a small population of ACs that invade).

Supplementary tables:

Tables S2 and S3 is missing the plasmids and primers used to generate the egl-43Si and egl-43Li RNAi constructs. Information regarding the targeting sequences used would also be helpful to include.

Table S2 contains primers whose sequences are not provided in Table S3. Namely oTD140-143.

Table S3 contains sequences of primers that are not defined in Table S2 or elsewhere. Namely the OEL316-319.

Reviewer #3: The C. elegans AC-VU cell fate decision is a classic example of Notch-based lateral inhibition, and the two fate outcomes differ in proliferation and invasive (basement membrane breaching) activity. In this manuscript, the authors follow up on prior studies from >10 years ago that showed that EGL-43, an Evi1-related transcription factor, is required for both the VU cell fate and for AC invasion. They focus particularly on EGL-43 function during AC invasion.

Key findings:

-The long isoform of egl-43 is required for AC invasion despite unimportance of several specific protein domains unique to this isoform (based on RNAi, AC-specific deletion, and CRISPR-generated in-frame deletions).

-EGL-43 inhibits AC proliferation by promoting G1 arrest, and does so by (directly or indirectly) repressing LIN-12/Notch, which promotes proliferation and inhibits invasion (based on increasing AC numbers in egl-43i over time, with increased expression of LIN-12::GFP and a variety of S-phase reporters, and epistasis analysis with lin-12 rf and gf). This is the most impactful result in the paper.

-EGL-43 does not affect expression of two other factors involved in AC invasion (NHR-67 and HDA-1), or vice versa, but EGL-43 and FOS-1 mutually upregulate each others’ expression

Overview:

The paper is generally well written, though data presentation needs clarification in several places. The presented results seem solid and the paper definitely moves the field forward, but it stops short of a really impactful, mechanistic understanding. Therefore, my enthusiasm for the current version is modest.

Specific points:

1. There are some “rigor and reproducibility” issues in the data presentation that need to be addressed:

Fig 1B quantification: how many animals were examined, how many showed these expression patterns, and how many had multiple ACs and/or no invasion?

Fig 2B: statistics? Are the changes over time significant?

Fig 2C and 2D quantification: how many animals were examined, how many showed these expression patterns?

Figure 3H and 3I: What % of animals had multiple ACs? How does this relate to the % that had invasion defects?

Figure 4F and 4G: Are the %s in G based on pooling the 3 lines of each genotype, or are they based on one specific line? What % of the animals scored in G had multiple ACs and is there an absolute correlation between the two types of defects?

Addressing any of the below would add more meat and impact to this paper, but addressing points 2,3,4 would be especially relevant:

2. Circumstantial evidence suggests that the egl-43 long isoform might be required because it is the predominant one transcribed in the AC, but the authors can't exclude that there is something unique about the protein made by this isoform. Transgenic rescue experiments with each isoform could address this question more definitively.

3. The authors present more evidence for the previously observed correlation between cell cycle arrest and invasive activity, but the basis for this correlation remains unclear. On p. 10 the authors speculate that "EGL-43 might perform two functions in the AC; first to repress AC proliferation and second to activate pro-invasive gene expression". But data in Figure 4 suggest that EGL-43 might do one thing (inhibit lin-12) to accomplish both results. What is the explanation for why lin-12(gf) suppresses proliferation and invasion defects (Figure 4), but CKI expression mostly does not (Figure 3)? At what step of the cell cycle is lin-12 acting? Does lin-12 affect expression of the mentioned "pro-invasive" genes?

4. Is the cross-regulation of egl-43 and fos-1 functionally important? Can cdh-3>fos-1 rescue egl-43 defects or vice versa? Or are they both needed in parallel?

5. The paper does not show whether lin-12, fos-1, or any of the other downstream effectors of EGL-43 are direct targets, so it does not provide insights into EGL-43 binding properties or mechanisms of transcriptional regulation (e.g. is EGL-43 both a tx repressor and activator?).

6. Given that egl-43 is expressed in both AC and VU, and seems to regulate different target genes in each, what is the basis for target specificity?

**Have all data underlying the figures and results presented in the manuscript been provided?**

Reviewer #1: Yes

Reviewer #2: No: We cannot find information pertaining to the sequence of the egl-43Si 5'UTR used in the generation of the RNAi targeting clone to the short isoform of egl-43. Also, I could not find the data supporting the claim that the different GFP-tagged alleles have the same expression levels in the AC - Fig. S2 should use the unmodified allele to normalize data to.

Reviewer #3: No: numerical data for bar graphs has not been provided

PLOS authors have the option to publish the peer review history of their article (what does this mean?). If published, this will include your full peer review and any attached files.

Reviewer #1: No

Reviewer #2: Yes: David Matus

Reviewer #3: No

---

## [Decision Letter · Decision Letter 1]

17 Feb 2020

Dear Dr. Hajnal

Thank you very much for submitting your Research Article entitled 'The C. elegans homolog of the Evi1 proto-oncogene, egl-43, coordinates G1 cell cycle arrest with pro-invasive gene expression during anchor cell invasion' to PLOS Genetics. Your manuscript was fully evaluated at the editorial level and by independent peer reviewers. The reviewers appreciated the attention to an important topic but identified some aspects of the manuscript that should be improved.

After carefully reading through Reviewer#1's comments (self identified as David Matus), I think he brings up a reasonable editorial suggestion in improving the manuscript--more strongly suggesting an alternative model that EGL-43 might also function to repress the cell cycle in part by promoting cki-1 expression.  I don't think you need to bring up the ChIP data, but I leave that up to you.  If you can make this change and outline how you have done so in your letter, I will make sure this elegant work is accepted quickly--it will not be sent out for re-review.

Upload a Striking Image with a corresponding caption to accompany your manuscript if one is available (either a new image or an existing one from within your manuscript). If this image is judged to be suitable, it may be featured on our website. Images should ideally be high resolution, eye-catching, single panel square images. For examples, please browse our archive. If your image is from someone other than yourself, please ensure that the artist has read and agreed to the terms and conditions of the Creative Commons Attribution License. Note: we cannot publish copyrighted images.

[LINK]

Yours sincerely,

David R Sherwood

Guest Editor

PLOS Genetics

Gregory P. Copenhaver

Editor-in-Chief

PLOS Genetics

Reviewer's Responses to Questions

**Comments to the Authors:**

Reviewer #2: Summary:

I commend the authors on the extensive revisions they performed to address many of the comments provided by myself and other reviewers previously. Both the text and graphs are much clearer now. The only result that we’d like to bring attention to is still the discrepancy in data between this report and recent work from our laboratory. We are not asking for additional experiments at this time, but would just like to offer some caveats that might change the interpretation of the data, regarding the role of EGL-43 in maintaining the post-mitotic state of the AC independent of the activity of CKI-1.

In 2015 we reported that an integrated transgene (cdh-3>CKI-1::GFP) completely rescued the AC proliferation and invasion defects in the hypomorph, nhr-67(pf88). In our 2020 paper we demonstrated this same transgene also completely rescued invasion and proliferation in nhr-67(RNAi) using the enhanced RNAi vector in the T444T backbone. In Figure 3F and 3G it appears that your extrachromosomal cdh-3>CKI-1::SL2::mNG transgene only partially rescues nhr-67(RNAi) [>30% invasion defect, and >10% with 2+ ACs]. Both the use of extrachromosomal arrays, which can vary in expression levels and the construction of your transgene using an SL2::mNG may explain the difference between our results and your results presented in Figure 3. Specifically, Ahier and Jarriault (Genetics 2014, doi: 10.1534/genetics.113.160846) argue for using 2A viral peptides rather than SL2 sequences in generating fusion proteins where stoichiometry between the protein of interest and the fluorescent protein are important to maintain. Here’s the relevant text from their discussion section:

“This is in contrast to the SL2 approach as operon sequences may be under the influence of cis-regulating elements (enhancers or silencers) that are difficult to predict (Pfleger et al. 2006). Moreover, when using an SL2-based approach, several additional parameters can impact on the expression levels of protein products, which could vary widely. For example, after trans-splicing, each subsequent mRNA molecule may have its own specific stability, and each transcript is read individually by different ribosomes, permitting variable translation.”

In our integrated transgene, cdh-3>CKI-1::GFP, we avoid this by generating the fusion between CKI-1 and GFP.

*If* the extrachromosomal lines more robustly rescued nhr-67(RNAi), I would be inclined to believe that egl-43 could be functioning independently of CKI-1 through LIN-12, but the more parsimonious explanation is that you’re not generating enough CKI-1 to restore the AC to a G0 cell cycle arrested state. I think this only affects the summary model - that the repression of LIN-12 is required with NHR-67 to maintain G0 arrest is an amazing result. I think that we don’t know (yet) whether LIN-12 functions to either activate cyclins and CDK complexes or repress CKI-1. I think this is made even more complex because of the likely feedback between CKIs and CDK/Cyclin complexes. I would be fine with showing dotted arrows with a question mark between LIN-12 and the cell cycle machinery, to show that the current data can’t resolve this relationship.

Finally, I also noticed that cki-1 might actually be a direct target of egl-43, from your L3 ChIP data presented in the excel file. This is exciting, but obviously potentially counter to your proposed model. It suggests that nhr-67 and egl-43 might co-regulate CKI-1 activity in the AC to maintain it in a post-mitotic G0 arrested state, though we still don’t know if nhr-67 directly regulates CKI-1.

chrII:7810217-7810482 cki-1; cki-1 356.7129 15.22800064

chrII:7809138-7809378 cki-1 170.3209 6.874500275

Reviewer #3: The authors have responded satisfactorily to my prior comments, and the revisions have significantly improved the paper.

**Have all data underlying the figures and results presented in the manuscript been provided?**

Reviewer #2: Yes

Reviewer #3: Yes

PLOS authors have the option to publish the peer review history of their article (what does this mean?). If published, this will include your full peer review and any attached files.

Reviewer #2: Yes: David Matus

Reviewer #3: No

---

## [Editor Report · Decision Letter 2]

27 Feb 2020

Dear Dr Hajnal

We are pleased to inform you that your manuscript entitled "The C. elegans homolog of the Evi1 proto-oncogene, egl-43, coordinates G1 cell cycle arrest with pro-invasive gene expression during anchor cell invasion" has been editorially accepted for publication in PLOS Genetics. Congratulations!

Yours sincerely,

David R Sherwood

Guest Editor

PLOS Genetics

Gregory P. Copenhaver

Editor-in-Chief

PLOS Genetics

**Data Deposition**

http://datadryad.org/submit?journalID=pgenetics&manu=PGENETICS-D-19-01671R2

**Press Queries**

---

## [Editor Report · Acceptance letter]

12 Mar 2020

PGENETICS-D-19-01671R2 

The *Caenorhabditis elegans* homolog of the *Evi1* proto-oncogene, *egl-43*, coordinates G1 cell cycle arrest with pro-invasive gene expression during anchor cell invasion 

Dear Dr Hajnal, 

We are pleased to inform you that your manuscript entitled "The *Caenorhabditis elegans* homolog of the *Evi1* proto-oncogene, *egl-43*, coordinates G1 cell cycle arrest with pro-invasive gene expression during anchor cell invasion" has been formally accepted for publication in PLOS Genetics! Your manuscript is now with our production department and you will be notified of the publication date in due course.

With kind regards,

Matt Lyles

PLOS Genetics

On behalf of:
